# Changes in the Subchondral Bone, Visfatin, and Cartilage Biomarkers after Pharmacological Treatment of Experimental Osteoarthritis with Metformin and Alendronate

**DOI:** 10.3390/ijms241210103

**Published:** 2023-06-14

**Authors:** Sevdalina Nikolova Lambova, Nina Ivanovska, Stela Stoyanova, Lyudmila Belenska-Todorova, Elenka Georgieva, Tsvetelina Batsalova, Dzhemal Moten, Desislava Apostolova, Balik Dzhambazov

**Affiliations:** 1Department of Propaedeutics of Internal Diseases, Faculty of Medicine, Medical University—Plovdiv, 4000 Plovdiv, Bulgaria; sevdalina_n@abv.bg; 2Department of Immunology, The Stephan Angeloff Institute of Microbiology, Bulgarian Academy of Sciences, 1113 Sofia, Bulgaria; nina_ivanovska@abv.bg; 3Department of Developmental Biology, Paisii Hilendarski University of Plovdiv, 4000 Plovdiv, Bulgaria; stela.stoyanova@uni-plovdiv.bg (S.S.); elenkageorgieva@uni-plovdiv.bg (E.G.); tsvetelina@uni-plovdiv.bg (T.B.); moten@uni-plovdiv.bg (D.M.); apostolova@uni-plovdiv.bg (D.A.); 4Faculty of Medicine, Sofia University “St. Kliment Ohridski”, 1407 Sofia, Bulgaria; lbelenska@uni-sofia.bg

**Keywords:** osteoarthritis, metformin, alendronate, biomarkers, visfatin, COMP, MMP-13, CTX-II, histopathology

## Abstract

Subchondral bone that has intense communication with the articular cartilage might be a potential target for pharmacological treatment in the early stages of osteoarthritis (OA). Considering the emerging data about the role of adipokines in the pathogenesis of OA, the administration of drugs that influence their level is also intriguing. Metformin and alendronate were administered in mice with collagenase-induced OA (CIOA) as a monotherapy and in combination. Safranin O staining was used for the assessment of changes in subchondral bone and articular cartilage. Before and after treatment, serum levels of visfatin and biomarkers of cartilage turnover (CTX-II, MMP-13, and COMP) were assessed. In the current study, the combined administration of alendronate and metformin in mice with CIOA led to the protection against cartilage and subchondral bone damage. In mice with CIOA, metformin led to a decrease in visfatin level. In addition, treatment with metformin, alendronate, or their combination lowered the level of cartilage biomarkers (CTX-II and COMP), while the level of MMP-13 was not influenced. In conclusion, personalized combination treatment in OA according to clinical phenotype, especially in the early stages of the disease, might lead to the identification of a successful disease-modifying therapeutic protocol in OA.

## 1. Introduction

Osteoarthritis (OA) is a chronic joint disease affecting a significant part of the human population and leading to physical disability and impaired quality of life [1]. OA is a heterogeneous and a multifactorial disorder characterized by degeneration, but also by low-grade inflammation [2,3]. OA involves impairment in the entire joint, including cartilage loss; subchondral bone remodeling; the formation of osteophytes; the development of bone marrow lesions; changes in the joint capsule, synovium, ligaments, and periarticular muscles; and meniscal tears and extrusion [4]. More recently, obesity and metabolic syndrome have been suggested as contributing factors to the pathological process in OA [2,3].

Cartilage destruction correlates with subchondral bone changes in the early stages of OA [5]. Subsequently, in the late chronic stages of the disease, a reduction in bone remodeling and amplified bone densification result in bone sclerosis [6]. Articular cartilage contains chondrocytes and extracellular matrix (ECM) composed of proteoglycans and collagens, and a small amount of glycoproteins, elastin, and gelatin [7].

A main feature of OA is considered to be the articular cartilage destruction. Therefore, the products of this process have been preferred biomarkers for analyzing the stage and prognosis of the disease [8]. Articular cartilage is subjected to metabolic turnover maintained mainly by collagen-degrading enzymes of the family of matrix metalloproteinases (MMPs) and their inhibitors [9]. MMP-13 (collagenase 3) is a key factor for collagen type II (COL2) cleavage and cartilage destruction in OA [10]. Cartilage oligomeric matrix protein (COMP), also known as thrombospondin 5, is a key factor for collagen assembly and extracellular matrix stability [11]. COMP is suggested to be a reliable serum and synovial fluid marker that reflects the process of articular destruction in OA and can be used for monitoring the response to different therapeutic agents [12,13]. Another cartilage degradation marker is C-terminal cross-linked telopeptide of type II collagen (CTX-II) [14]. It is a byproduct of COL2 breakdown that is located almost exclusively in the cartilage [15]. According to the meta-analyses of recent reviews, COMP and CTX-II have been suggested as appropriate biomarkers in predicting disease progression in knee OA [16,17].

Obesity affects a significant part of the human population. The biomechanical overload in obese individuals is associated with the development of knee OA. However, it has been observed that obesity is also associated with hand OA that suggests the role of systemic factors [18] such as adipokines produced by the adipose tissue [19]. The observation of differences in clinical and laboratory characteristics in patients with knee OA with and without obesity led to the novel suggestion of the existence of a distinct subtype of metabolic knee OA [20,21]. Visfatin is a novel adipokine that is suggested to play a detrimental role in cartilage and bone metabolism in OA, as it increases the production of MMPs, tumor necrosis factor-α (TNF-α), interleukin (IL)-1β, and IL-6, and induces osteophyte formation by inhibiting osteoclastogenesis [22,23]. The current treatment of OA is constantly evolving and exploring novel therapeutic strategies that aim to target not only the symptoms and phenotype characteristics of the disease, but also its complex mechanisms [24,25,26]. A significant challenge is the development of therapeutic strategies that not only reduce pain, but also have the potential to slow down or even prevent destruction of the articular cartilage and changes in the subchondral bone. Despite the presence of low-grade inflammation in OA, the use of disease-modifying drugs that are established for treatment of inflammatory arthritides is disappointing in patients with OA [27]. Currently, there are no approved disease-modifying treatments for OA that are related to multiple factors such as heterogeneity in disease pathogenesis and the existence of different clinical phenotypes. Subchondral bone remodeling is a key feature of OA that is associated with articular cartilage destruction and leads to bone sclerosis and irreversible damage [6]. Bisphosphonates appear to be a promising option for OA treatment as they inhibit bone resorption and are suggested to have a positive impact on articular cartilage and periarticular bone changes [28]. Bisphosphonates are indicated for treatment of osteoporosis due to their inhibitory effects on the osteoclast-mediated bone resorption [29]. Alendronate is a nitrogen-containing bisphosphonate and potent inhibitor of osteoclastic resorption that has demonstrated positive results in preclinical models of OA after systemic and intraarticular administration [30,31].

Metformin (1,1-dimethylbiguanide) is a member of the biguanide class of compounds and used as a first-line therapy for diabetes type 2 combined with obesity [32]. A summary of the recent data obtained from in vitro studies, animal models, and clinical trials describes that metformin can slow OA progression due to its anti-inflammatory effects (decrease in the level of proinflammatory cytokines and leptin), anti-oxidative effects, upregulation of autophagy, and modulation of the microbiome [33]. Metformin was suggested to suppress osteoclastogenesis and to diminish RANKL expression in osteoblasts [34].

Previously, we have examined the disease-modifying potential of metformin, alendronate, and their combination in an experimental mouse model of osteoarthritis [35]. We suggested that the combination of alendronate and metformin might protect the excessive bone resorption; preserve bone integrity, structure, and function; and also influence the metabolic aspects in the osteoarthritic process.

Collagenase-induced osteoarthritis (CIOA) is an animal model, where the disease is induced by the intra-articular injection (i.a.) of collagenase, and some of the OA-like changes that are typical in humans can be observed. It is suitable for studying the histopathological and biochemical changes during cartilage and bone remodeling. It was previously found that arthritis in ICR mice resemble most of the characteristics typical of human pathology [36]. In the present study, we aimed to investigate the effect of metformin, alendronate, and their combination on serum levels of the adipocytokine visfatin, as well as on the expression of markers for cartilage degradation such as MMP-13, COMP, and CTX-II. We expanded our previous study of metformin, alendronate, and their combination on the reduction in the histopathological and biochemical signs of OA, suggesting their positive effect on the disease progression. Nevertheless, there is an urgent need to develop rapid and non-invasive methods for assessment of the current state of patients’ joints, with which to monitor the ongoing treatment.

## 2. Results

### 2.1. Influence of Metformin and Alendronate on the Subchondral Bone Changes

Histopathological changes in the subchondral bone of the experimental animals are shown in Figure 1 and Table 1. Our results showed that the articular cartilage of the healthy control (Group 1) is directly connected to bone marrow via open fenestrae in the subchondral plate (Figure 1A). The mice with CIOA were characterized by severe sclerosis (Table 1). A massive increase in subchondral bone volume was found (Figure 1B). A second degree of histopathological changes in experimental group 3 (CIOA + metformin) and group 4 (CIOA + alendronate) was observed (Figure 1C,D). These alterations showed an increase in subchondral bone sclerosis, as well as the lack of an open connection between bone marrow and cartilage. In the experimental animals of Group 5 (CIOA + metformin + alendronate), we found subchondral sclerosis and an increase in bone volume in a mild degree of expression (Figure 1E). Moreover, thickened subchondral bone trabeculae were observed.

### 2.2. Influence of Metformin and Alendronate on the Serum Levels of Visfatin, CTX-II, MPP-13, and COMP

The levels of different serum markers were determined on day 30 of CIOA. Arthritic mice showed an elevation of visfatin in the chronic phase of joint inflammation, which was lowered in the groups treated with metformin alone or in combination with alendronate (Figure 2A). Treatment with alendronate alone showed no changes in serum visfatin levels compared to the untreated CIOA mice (Figure 2A). CTX-II was another marker expressed higher in the sera of the CIOA group successfully inhibited by metformin, alendronate, or their combination (Figure 2B). With regard to the MPP-13 level, no changes were observed in non-treated or treated CIOA mice (Figure 2C). The arthritis process was related to an increased serum COMP level, which was significantly reduced by both substances and their combination (Figure 2D).

### 2.3. Serum Levels of Visfatin in Patients with Knee OA

In patients with knee OA, serum levels of visfatin (22.33 ± 15.66 ng/mL) were higher as compared with the control group, which comprised individuals with different BMIs, including cases with obesity but without OA (17.840 ± 8.563 ng/mL), but the difference was not statistically significant (*p* < 0.16) (Figure 3).

In control subjects, visfatin levels correlated positively with BMI (*p* < 0.05). There was no such correlation in knee OA patients (Figure 4). An association between visfatin concentration with age and sex was not found in both patients and controls.

## 3. Discussion

Osteoarthritis is related to progressive changes in the cartilage, bone, and synovium, resulting in painful articular cartilage loss. 

### 3.1. Subchondral Bone as a Therapeutic Target

The subchondral bone grading scale proposed by Aho et al. [37] is focused on bone remodeling, and the increase in bone volume is perhaps the most important change in higher grades. In addition, early OA changes in subchondral bone are thought to be caused by increased bone remodeling and an increase in remodeling sites in bone, further leading to reduced subchondral plate thickness [38]. Regarding the regressive joint changes, we observed severe histopathological changes in the subchondral bone of CIOA mice associated with severe sclerosis and a massive increase in subchondral bone volume. These changes showed that we found a link between increased bone remodeling and cartilage degeneration. Furthermore, according to Aho et al. [37], no association between bone marrow and cartilage could be observed. In groups 3 and 4, we found grade 2, which was associated with a distinct increase in subchondral bone sclerosis and volume. No connection between bone marrow and cartilage could be observed either. In contrast to groups 3 and 4, in the mice treated with CIOA + metformin + alendronate, we observed a mild expression of subchondral bone changes, which we evaluated as grade 1. According to the proposed scale, we observed thickened subchondral bone trabeculae, and contact between the bone marrow and cartilage could be seen.

Alendronate expresses the ability to maintain bone integrity [39] and to inhibit subchondral bone remodeling in animal models of OA. The chondrocyte protection and inhibition of osteophyte formation have also been detected after treatment with alendronate. The mechanism of chondroprotection realized by alendronate is not fully understood. Although possible direct and indirect effects of bisphosphonates on cartilage have been suggested, the most probable mechanism might be the inhibition of subchondral bone turnover [40].

Fernández-Martín et al. [41] performed a systematic review of the potential of bisphosphonate as disease-modifying agents in preclinical models of OA based on the data of 26 publications with five different types of bisphosphonates. The authors concluded that treatment with bisphosphonates is associated with the preservation of subchondral bone and fewer biomarker changes. Suppression of the osteophyte development was not confirmed and a high variation in their chondroprotective effect was registered in different studies. An anti-inflammatory effect of bisphosphonates on the synovial membrane was also suggested. An important point to consider is the time of treatment initiation according to the OA stage. Beneficial effects of bisphosphonates are better when they are initiated early in the disease course that could be applied in preclinical models [41], while in humans with OA, early stages of the disease might be subclinical or asymptomatic, thus excluding the opportunity for early intervention. In a meta-analysis of Vaysbrot et al. [42], the efficacy of bisphosphonates in knee OA was studied based on the analysis of the available data from clinical studies. The conclusion was that bisphosphonates do not provide symptomatic relief and do not retard the radiographic progression in knee OA. However, beneficial effects from bisphosphonate treatment could be expected in subsets of patients who exhibit high rates of subchondral bone turnover [42]. Moreover, treatment with the bisphosphonate zoledronic acid has shown a reduction in the size of bone marrow lesions in a placebo-controlled trial in patients with knee OA (*n* = 59) [43]. Bone marrow lesions/edema are located in the subchondral bone and may predict structural progression, symptom activation, and pain in patients with knee OA. These lesions are frequently found in conjunction with cartilage damage in the same zone of the knee. Incident bone marrow lesions and an increase in their size over time predict future cartilage loss. At histological examination, bone marrow lesions represent areas of subchondral bone damage and remodeling, and changes indicative of fibrosis, necrosis, and trabecular alterations could be found in the affected bony areas [44]. However, conflicting results are reported regarding the effect of bisphosphonates on bone marrow lesions in knee OA. In a multicenter, double-blind placebo-controlled randomized clinical trial with a 24-month duration that included 223 participants, 190 of whom completed the trial, intravenous infusion with 5 mg of zoledronic acid was associated with a significant change neither in knee pain assessed by a visual analog scale and Western Ontario and McMaster Universities Osteoarthritis, nor in bone marrow lesion size and cartilage volume loss [45].

### 3.2. Visfatin and Cartilage Biomarkers as Potential Targets for the Pharmacological Treatment in Osteoarthritis

There is a great need to identify sensitive biomarkers to predict the initiation and progression of OA. Much attention has been focused on finding new molecular markers, particularly cartilage-derived fragments whose level in the circulation may reflect disturbances in joint tissue repair. Visfatin is a relatively new adipokine that has generated increasing interest. It has been reported that high visfatin concentrations in rheumatoid arthritis patients correlate with the persistence of inflammation [46]. Visfatin induces the production of IL-1β, TNF-α, IL-6, and MMPs. Moreover, it is capable of increasing the surface expression of co-stimulatory molecules, such as CD54, CD40, and CD80, and it also enhances B-cell proliferation and inhibits neutrophil apoptosis in rheumatoid arthritis [47]. Some of the suggested biomarkers are pointed out to quantify joint remodeling associated with collagen metabolism in cartilage and bone. They may reflect joint tissue erosion or tissue repair and may be detected in synovial fluid, blood, or urine. Among the promising candidates are serum CTX-II and COMP. Type II collagen and aggrecan represent the most characteristic proteins in the cartilage matrix [48,49]; particularly, type II collagen represents 80–90% of the collagen in cartilage tissue. MMPs are the main players in the degradation of type II collagen as a hallmark of irreversible cartilage damage [15,50]. COMP prevails in the articular cartilage compared to other joint tissues so its serum levels might be representative of cartilage catabolism [51]. COMP plays a pathogenetic role in RA and OA that probably is independent of the mechanisms regulating persisting inflammatory processes [52,53]. Higher serum COMP levels in OA patients indicate the potential ability of COMP to predict disease severity. Metformin was first introduced as an antidiabetic drug, but data later revealed that it was able to suppress osteoclastogenesis and diminish RANKL expression in osteoblasts [34]. Alendronate expresses the ability to maintain bone integrity in several animal models [39], except alendronate realizes chondrocyte protection and inhibition of osteophyte formation [40].

Our results showed that in the chronic phase of CIOA, the increased visfatin level in the circulation was less expressed in the group with OA, treated with metformin as a monotherapy and the combination of metformin and alendronate. The increased serum levels of CTX-II and COMP were significantly lowered after treatment with metformin, alendronate as a monotherapy, and their combination. Alendronate is associated with decreases in the level of biomarkers of cartilage and bone metabolism. In contrast, MMP-13 was not affected by CIOA development and the administered treatment. It should be noticed that the results are associated only with the late phase of CIOA. In this context, there is uncertainty in defining whether MMP-13 is not a reliable marker at all or is in the chronic phase. 

The results suggest that metformin could influence the level of visfatin and the markers of cartilage turnover. A decrease in visfatin levels after metformin administration was also reported in a high-fat-diet-induced obesity model in rats [54]. In humans, metformin administration has also led to a decrease in visfatin levels in patients with polycystic ovary syndrome after 3 months [55]. Metformin treatment for 8 weeks in adolescents with obesity has also been associated with a significant decrease in the level of other key adipokine—leptin [56].

Concomitant administration of metformin and alendronate does express a synergistic effect. However, their use in combination aims to target different pathways in the complex pathogenesis of OA. The first report about the successful influence of different aspects of disease pathogenesis in CIOS by combined administration of metformin and alendronate was reported by our group in 2021 [35].

In patients with knee OA, serum levels of visfatin were higher as compared with the control group, which comprised individuals with different BMIs, including cases with obesity but without OA, although the difference between the patients and controls was not statistically significant. The higher values of visfatin in patients with knee OA could be related to the higher BMI of the patients in our study or with the osteoarthritic process. In the current study, a positive correlation between visfatin levels and BMI was found only for controls. Visfatin is suggested to be involved in OA pathogenesis. Its level has been assessed in human OA knee explants and synovial fluids in patients undergoing total joint replacement. It was confirmed that visfatin is released by all OA tissues, i.e., the synovium, subchondral bone, and cartilage. The release of visfatin was significantly higher in the synovial membrane vs. Articular cartilage (*p* = 0.0003) and subchondral bone (*p* = 0.0012), without a significant difference between OA cartilage and subchondral bone (*p* = 0.08). Visfatin was also detected in synovial fluid [57]. In this regard, further analysis of the association of visfatin levels with the stage and activity of knee OA is worth further attention.

In addition, the data in the current study prove the usefulness of CIOA as a model for identifying appropriate markers relevant to the initiation and severity of the osteoarthritic process. This model allows also evaluation of the therapeutic effect of pharmacological treatment.

## 4. Materials and Methods

### 4.1. Animals, Induction of OA, and Treatment

Female ICR (CD-2) mice, 8–10 weeks of age, with average body weights of 20–24 g, were used in the experiments. They were fed with standard rodent chow and water *ad libitum*. A total of 50 mice were divided into five groups (10 mice per group). The animals were fed with standard pelleted chow and tap water ad libitum.

To induce OA, 40 mice (four OA experimental groups) were injected (i.a.) with 2U of collagenase from *Clostridium histolyticum* (Sigma-Aldrich, Saint Louis, MO, USA) in a total volume of 10 μL under anesthesia (sodium pentobarbital 50 mg/kg, i.p.). The control group mice were injected (i.a.) with 10 μL of phosphate-buffered saline (PBS) 10 times (every other day from day 0 to day 18).

The experimental design and metformin/alendronate treatments were the same as previously described [35]. Briefly, one of the OA experimental groups was treated (i.p.) 10 times with PBS (every other day from the day of i.a. injection of collagenase to day 18). Similarly, the other three OA groups were treated 10 times with metformin (i.p. 10 mg/kg), alendronate (i.p. 20 mg/kg), or metformin plus alendronate (i.p. 10 mg/kg each).

Alendronate is approved for oral administration in humans for the treatment of osteoporosis. The bioavailability after oral administration is low (0.9 to 1.8%) and food inhibits the oral absorption [58]. Intraperitoneal administration of alendronate is an established approach in animal models [59,60] with radiologically confirmed increased bone density in mice as demonstrated by Monma et al. [59].

All animal procedures were performed in accordance with the Guidelines of the Bulgarian Food Safety Agency (Protocol No 352/6 January 2012) and the international laws and policies (EEC Directive of 1986; 86/609/EEC, recommendation 2007/526/EC from European Community), and allowed by the Animal Ethical Committee at the Institute of Microbiology, Bulgaria (protocol No 23, 18 May 2020). 

### 4.2. Histopathological Analysis

The Safranin O staining protocol was applied, and all histological samples for light microscopic analysis were prepared according to Schmitz et al. [61]. All knee joints were fixed for 3 days in 10% buffered paraformaldehyde (pH 7.0), decalcified by incubation in 10% ethylenedia-minetetraacetic acid (EDTA) (in 0.1 M phosphate buffer, pH 7 for approximately 2 weeks, solution changed four times a week) [62], cleared with xylene, infiltrated with liquid paraffin with a melting point of 54–56 °C, and finally enclosed in paraffin. Thus, sections of 4 μm were prepared and stained with Safranin O and Fast green before they were further studied with a light microscope (Leica DM 2000 LED, Leica Microsystems, Wetzlar, Germany) connected to a digital camera (Leica DM 2000 LED, Leica Microsystems, Wetzlar, Germany). A four-degree (0–3) severity gradation scale was applied and the histological alterations of the subchondral bone were characterized according to Aho et al. [37].

### 4.3. ELISA Assays

Blood was collected by retro-orbital puncture and was allowed to clot for 1 h at room temperature in order to separate sera. The levels of different markers were quantified by the following kits: mouse visfatin ELISA kit (sensitivity, 0.375 ng/mL), mouse CTX-II ELISA kit (sensitivity, 0.78 pg/mL), mouse MMP-13 ELISA kit (sensitivity, 0.11 ng/mL), and mouse Comp ELISA kit (sensitivity, 0.938 ng/mL), all from MyBioSourse (San Diego, CA, USA).

### 4.4. Patients

A total of 135 patients with symptomatic primary knee OA at the age between 35 and 88 years (mean age, 66 years) were included in the study (119 women and 16 men). A total of 34 individuals including patients with obesity but without radiographic knee OA (Kellgren–Lawrence = 0) were examined as a control group. The control group was significantly younger and with a lower BMI than the patients with knee OA. This is related to the high frequency of radiographic knee OA in elderly albeit asymptomatic individuals. All patients underwent clinical examination and antero-posterior radiologic examination of knee joints bilaterally in a standing position. The inclusion criterion for participation in the study was the presence of symptomatic and radiologically confirmed primary knee OA ≥ 2nd radiographic stage according to the Kellgren–Lawrence scale [63]. Exclusion criteria included the presence of different causes for secondary knee OA, i.e., posttraumatic knee OA, congenital or developmental diseases—either localized or generalized—inflammatory joint disease (rheumatoid arthritis, reactive arthritis, etc.), gout or other crystal arthropathies, history of septic arthritis, and bone pathology or other diseases that could be associated with secondary knee OA [64]. All patients had bilateral knee involvement. The patients were from the 2nd to 4th radiographic stage according to the Kellgren–Lawrence scale. Patients, in whom the radiologic stage differed at the two knees, were classified into the more severe degree group. A total of 75 patients were with concomitant obesity (BMI ≥ 30 kg/m^2^) and 60 patients had BMI < 30 kg/m^2^. Blood was drawn after 12 h fasting of the participants. Serum levels of visfatin were measured using a Human Visfatin ELISA Kit (MyBioSourse, San Diego, CA, USA) with a sensitivity of 0.23 ng/mL. All assays were performed according to the manufacturer’s instructions.

### 4.5. Statistical Analysis

Data are presented as mean ± standard deviation (SD). Statistical analyses were performed by one-way ANOVA, the unpaired *t*-test, and the Spearman rank correlation using InStat3.0 and GraphPad Software (La Jolla, CA, USA). Differences were considered to be significant at *p*  <  0.05.

## 5. Conclusions

Despite the new achievements in the basic science regarding the complex pathogenesis in OA, there is still no disease-modifying treatment for the disease. This is due mainly to the heterogeneity of the disorder and presence of different disease phenotypes and the avascularity of articular cartilage that does not permit adequate recovery, making the changes in this structure irreversible, especially in the advanced stages. Thus, an early personalized and complex intervention might eventually prevent or retard joint damage. In this regard, subchondral bone that has intense communication with the articular cartilage might be a potential target for pharmacological treatment in the early stages. Considering the emerging data about the role of adipokines in the pathogenesis of OA, the administration of drugs that influence their level is also intriguing. In the current study, the combined administration of the anti-resorptive agent alendronate and metformin in a mouse model of OA induced protection for cartilage and subchondral bone damage as compared with the control group. Metformin led to a decrease in visfatin levels. In addition, treatment with metformin, alendronate as a monotherapy, or their combination lowered the level of some cartilage biomarkers (CTX-II and COMP). In conclusion, personalized combination treatment in OA according to clinical phenotype, especially in the early stages of the disease, might lead to the identification of a successful disease-modifying therapeutic protocol in OA.

## Figures and Tables

**Figure 1 ijms-24-10103-f001:**
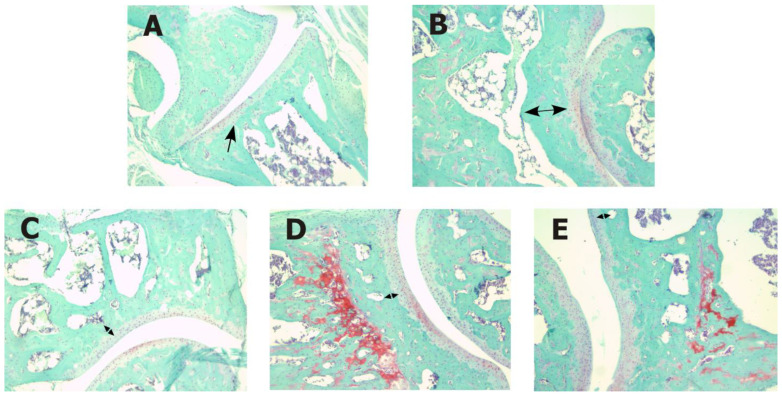
Effects of metformin and alendronate treatments on histopathological changes in the subchondral bone of CIOA in mice (Safranin O staining). (**A**) Section of normal (healthy control, *n* = 10) subchondral bone (Group 1, *n* = 10) with direct contact between cartilage and bone marrow (arrow). (**B**) Section of untreated CIOA mice (Group 2, *n* = 10) with a severe increase in the subchondral bone volume (double-headed arrow). Sections of CIOA mice treated with metformin (Group 3, *n* = 10) (**C**) or alendronate (Group 4, *n* = 10) (**D**) with a moderate increase in the subchondral bone volume (double-headed arrow). (**E**) Section of CIOA mice treated with metformin + alendronate (Group 5, *n* = 10) with a mild increase in the subchondral bone volume (double-headed arrow). Objective magnification: 40×.

**Figure 2 ijms-24-10103-f002:**
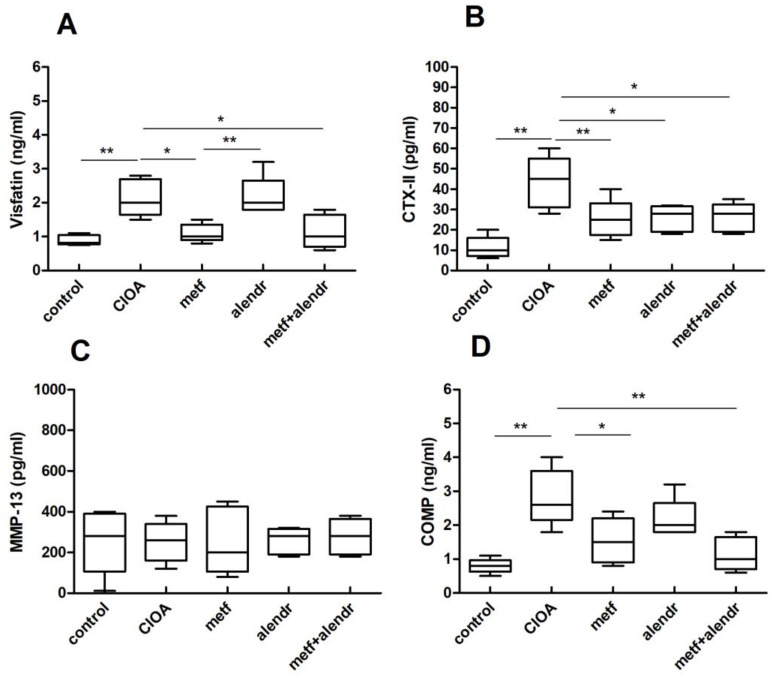
Serum visfatin (**A**), CTX-II (**B**), MMP-13 (**C**), and COMP (**D**) levels on day 30 of CIOA. Data are means ± SD from three determinations (*n* = 10 per group). * *p* < 0.05, ** *p* < 0.01 vs. CIOA group, unpaired *t*-test (CIOA = collagenase-induced osteoarthritis; metf = metformin; alendr = alendronate).

**Figure 3 ijms-24-10103-f003:**
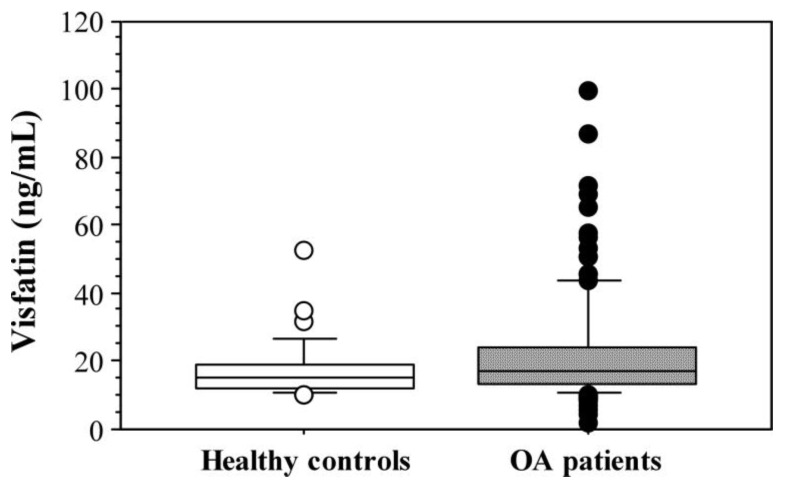
Serum visfatin levels in patients with primary knee OA (*n* = 135) and control subjects (*n* = 34).

**Figure 4 ijms-24-10103-f004:**
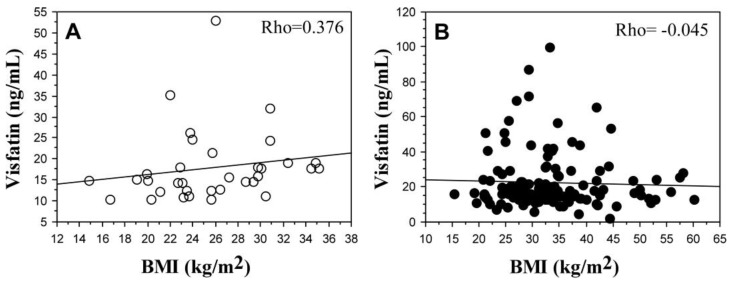
A correlation between visfatin serum levels and BMI. (**A**) Healthy controls (*n* = 34). (**B**) Knee OA patients (*n* = 135).

**Table 1 ijms-24-10103-t001:** Severity of the histopathological alterations in subchondral bone according to Aho et al. [37] grading system.

Experimental Group	Treatment	Degree of Histopathological Changes	Associated Criteria (Tissue Reaction)
Group 1	No treatment, healthy control	0	No subchondral sclerosis is observed and subchondral plate is thin. Articular cartilage is directly connected to bone marrow via open fenestrae in subchondral plate.
Group 2	No treatment, CIOA	3	Characterized by severe sclerosis and a massive increase in subchondral bone volume as the typical features. Loss of articular cartilage and flattened subchondral plate can be seen.
Group 3	CIOA + metformin	2	Characterized by a distinct increase in subchondral bone sclerosis and volume. Fibrillation in subchondral bone plate can be seen. No open connection between bone marrow and cartilage can be identified.
Group 4	CIOA + alendronate	2	Characterized by a distinct increase in subchondral bone sclerosis and volume. Fibrillation in subchondral bone plate can be seen. No open connection between bone marrow and cartilage can be identified.
Group 5	CIOA + metformin + alendronate	1	Characterized by subchondral sclerosis and an increase in bone volume. Open subchondral bone fenestrae connecting bone marrow to articular cartilage still exist. Thickened subchondral bone trabeculae can be seen.

## Data Availability

Data are contained within the article.

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
