# Peer review of "Changes in the Subchondral Bone, Visfatin, and Cartilage Biomarkers after Pharmacological Treatment of Experimental Osteoarthritis with Metformin and Alendronate"

_ijms, 2023, doi:10.3390/ijms241210103_

Round 1
Reviewer 1 Report
Interesting article.
A few remarks :
-the introdiction is too long
- how were alendronate and metformin administrated to mice (route of treatment). What were the doses.
- A clinical evaluation of mice could have been performed (functional impairment, joint pain, joint difformity)
- Was there a group of obese mice?
no comment
Author Response
Dear reviewers,
Thank you for your useful comments for improvement of our manuscript. Please find our answers below.
Reviewer 1
Comments and Suggestions for Authors
Interesting article.
A few remarks :
-the introdiction is too long
Response: The introduction has been revised and shortened.
- how were alendronate and metformin administrated to mice (route of treatment). What were the doses.
Response: Mice were treated 10 times with metformin (i.p. 10 mg/kg), alendronate (i.p. 20 mg/kg) or metformin plus alendronate (i.p. 10 mg/kg each), respectively. The details about route of administration and doses of alendronate and metformin were provided.
- A clinical evaluation of mice could have been performed (functional impairment, joint pain, joint difformity)
Response: Unfortunately, the CIOA animal model is not suitable for measurement of pain or joint deformity. This model is suitable to study biochemical and histopathological changes. The disease is evaluated for short period.
- Was there a group of obese mice?
Response: There was no group of obese mice. We do not have such animal model.
Reviewer 2 Report
The authors describe the effects of Metformin and alendronate in an experimental OA mouse model. This is an important topic that will continue to grow due to increasing life expectancy.
I have the following points that should be clarified.
Page 1, line 37-40: You state, that OA is a degenerative inflammatory joint disease and later you specify that ist is a low-grade inflammatory disease. That is an important point and it should be mentioned that an immunosuppressive approach is not a therapeutic option in this case, which could otherwise be assumed in an inflammatory disease (exception erosive OA, where anakinra treatment can be successful).
Page 2, line 68-74: You describe the role of obesity in the context of OA but you completely did not comment on the mechanical load due to obesity? Additional mechanical stress on the joint due to the additional weight should also be discussed in this context.
Figure description 1-3: The group sizes are given in Material and Methods, however, they should also be stated right away when labeling the figure: n=?? Especially important in the human data in Figure 3.
Page 7, line 214: the “I” is missing in anti-inflammatory.
Page 7, line 214-216: If bisphosphonates so exhibit an anti-inflammatory effect that might held to slow down the development of OA, are there any date that show less OA in patients being treated with BP due to osteoporosis?
Page 8, line 269: you write “arthritic mice” but you mean the OA mice? Arthritic mice could be mixed up with a model of rheumatoid arthritis?
Page 8, line 293-295: You state that the higher values of visfatin could be related to the higher BMI, so please show the date, you should have the patient characteristics, please show the correlation of the visfatin level and BMI/age/aney sex differences?
Page 9, line 320: does this mean that alendronate was also administered i.p.? How are you sure alendronate was absorbed?
Page 9, line 334 and following: Why did you not match the control patients (age/sex/BMI), the data would profit of matching?
English was fine to understand and follow.
Author Response
Dear reviewers,
Thank you for your useful comments for improvement of our manuscript. Please find our answers below.
Reviewer 2
The authors describe the effects of Metformin and alendronate in an experimental OA mouse model. This is an important topic that will continue to grow due to increasing life expectancy.
I have the following points that should be clarified.
Page 1, line 37-40: You state, that OA is a degenerative inflammatory joint disease and later you specify that ist is a low-grade inflammatory disease. That is an important point and it should be mentioned that an immunosuppressive approach is not a therapeutic option in this case, which could otherwise be assumed in an inflammatory disease (exception erosive OA, where anakinra treatment can be successful).
Response: The comment about the lack of efficacy of disease-modifying drugs in OA was added (Page 2, paragraph 4).
Page 2, line 68-74: You describe the role of obesity in the context of OA but you completely did not comment on the mechanical load due to obesity? Additional mechanical stress on the joint due to the additional weight should also be discussed in this context.
Response: The comment about the role of mechanical overload in obese patients was added (Page 2, paragraph 3).
Figure description 1-3: The group sizes are given in Material and Methods, however, they should also be stated right away when labeling the figure: n=?? Especially important in the human data in Figure 3.
Response: This information was added to the figure legends.
Page 7, line 214: the “I” is missing in anti-inflammatory.
Response: Corrected.
Page 7, line 214-216: If bisphosphonates so exhibit an anti-inflammatory effect that might held to slow down the development of OA, are there any date that show less OA in patients being treated with BP due to osteoporosis?
Response: The results are contradictory and major current data are discussed on page 7, paragraph 1 that include results from clinical trials with monitoring of radiographic progression and findings on magnetic resonance imaging.
Page 8, line 269: you write “arthritic mice” but you mean the OA mice? Arthritic mice could be mixed up with a model of rheumatoid arthritis?
Response: Corrected.
Page 8, line 293-295: You state that the higher values of visfatin could be related to the higher BMI, so please show the date, you should have the patient characteristics, please show the correlation of the visfatin level and BMI/age/aney sex differences?
Response: The data about correlation of visfatin with BMI, age and sex in patients and controls were added. We added Figure 4 related to the correlation between visfatin and BMI. Since there is no correlation with age and sex, we did not provide these figures, but this is written in the text.
Page 9, line 320: does this mean that alendronate was also administered i.p.? How are you sure alendronate was absorbed?
Response:
All the substances were administered i.p.
The intraperitoneal route of administration of alendronate has been established in previous animal models (Vieira et al. 2019, Monma et al. 2004 – a reference was added).
Data on the pharmacokinetics of alendronate are almost exclusively based on urinary excretion after oral or intravenous administration. It was accepted that about 60% of the dose persisted for a long time in the body, mainly in the skeleton (Porras, A.G.; Holland, S.D.;Gertz, B.J. Pharmacokinetics of Alendronate. Clinical Pharmacokineticss199, 36, 315–328, https://doi.org/10.2165/00003088-199936050-00002.
Page 9, line 334 and following: Why did you not match the control patients (age/sex/BMI), the data would profit of matching?
Response: The prevalence of osteoarthritis in elderly including subclinical forms is very high. For this reason, elderly controls without radiographic evidence of knee osteoarthritis are very rare.
Round 2
Reviewer 2 Report
Thank you, all points were clarified.